# Targeting the Microenvironment for Treating Multiple Myeloma

**DOI:** 10.3390/ijms23147627

**Published:** 2022-07-10

**Authors:** Peter Neumeister, Eduard Schulz, Katrin Pansy, Marta Szmyra, Alexander JA Deutsch

**Affiliations:** 1Division of Hematology, Medical University of Graz, Auenbruggerplatz 38, 8036 Graz, Austria; eduard.schulz@medunigraz.at (E.S.); katrin.pansy@medunigraz.at (K.P.); marta.szmyra@medunigraz.at (M.S.); alexander.deutsch@medunigraz.at (A.J.D.); 2Center for Cancer Research, National Cancer Institute, National Institutes of Health, Bethesda, MD 20892, USA

**Keywords:** microenvironment, multiple myeloma, immunology, targeted therapy, CD 38 antibody therapy, daratumumab, isatuximab, CAR T cell, bispecific antibody

## Abstract

Multiple myeloma (MM) is a malignant, incurable disease characterized by the expansion of monoclonal terminally differentiated plasma cells in the bone marrow. MM is consistently preceded by an asymptomatic monoclonal gammopathy of undetermined significance, and in the absence of myeloma defining events followed by a stage termed smoldering multiple myeloma (SMM), which finally progresses to active myeloma if signs of organ damage are present. The reciprocal interaction between tumor cells and the tumor microenvironment plays a crucial role in the development of MM and the establishment of a tumor-promoting stroma facilitates tumor growth and myeloma progression. Since myeloma cells depend on signals from the bone marrow microenvironment (BMME) for their survival, therapeutic interventions targeting the BMME are a novel and successful strategy for myeloma care. Here, we describe the complex interplay between myeloma cells and the cellular components of the BMME that is essential for MM development and progression. Finally, we present BMME modifying treatment options such as anti-CD38 based therapies, immunomodulatory drugs (IMiDs), CAR T-cell therapies, bispecific antibodies, and antibody-drug conjugates which have significantly improved the long-term outcome of myeloma patients, and thus represent novel therapeutic standards.

## 1. Introduction

Multiple myeloma (MM) is an incurable, heterogeneous malignancy characterized by malignant expansion of monoclonal terminally differentiated plasma cells in the bone marrow or more rarely, at extramedullary sites. In most patients, MM is characterized by the secretion of a monoclonal immunoglobulin (Ig), or Ig chain known as monoclonal protein or M-protein, which is produced by the abnormal plasma cell clone leading to the characteristic clinical manifestations of end-organ damage such as hypercalcemia (C), renal failure (R), anemia (A), and osteolytic bone destruction (B), collectively known as CRAB features [1,2]. The Revised International Myeloma Working Group (IMWG) diagnostic criteria for MM define active myeloma as clonal bone marrow (BM) plasma cell infiltration of ≥10% or a biopsy-proven bony or extramedullary plasmacytoma and at least one myeloma defining event such as the presence of a CRAB feature or presence of a biomarker associated with high-risk for end-organ damage. These biomarkers comprise of clonal bone marrow plasma cell infiltration ≥60%, involved/uninvolved serum free light chain (sFLC) ratio of ≥100, or >1 focal lesion of at least 5 mm in size on MRI studies.

Multiple myeloma belongs to a wide range of disorders referred to as monoclonal gammopathies. MM is consistently preceded by an asymptomatic monoclonal gammopathy of undetermined significance (MGUS), which is characterized by serum monoclonal protein (non-IgM type) <30 g/L, clonal bone marrow plasma cell infiltration <10%, and absence of end-organ damage (CRAB criteria). If the M-protein (IgG or IgA) is ≥30 g/L (or urinary M-protein ≥500 mg per 24 h) and/or the clonal bone marrow plasma cell infiltration reaches 10–60% in the absence of myeloma defining events the definition of smoldering multiple myeloma (SMM) is fulfilled [2] (Figure 1). Over 25 years, approximately 15% of patients with MGUS will progress to active MM. As mentioned, MGUS and SMM consistently precede MM with a 1% per year, life-long risk of progression to active myeloma in MGUS, and a 10% risk of progression in 5 years in SMM.

However, the molecular basis underlying malignant transformation from MGUS to active myeloma has not yet been entirely delineated. It is believed that malignant transformation is the consequence of clonal evolution of post-germinal center (GC) or plasma cells (PC) initiated by a primary genetic event and fueled by multiple accumulating secondary genetic events. Initiating events can be subcategorized into IgH-translocations or hyperdiploidy, whereas secondary genetic events responsible for progression are copy number variations, mutations, or epigenetic changes accumulating in the plasma cell clone. In the continuum of the distinct disease stages from MGUS to MM, progression is also promoted by a remodeling process exerted by various immune cells in the bone marrow microenvironment (BMME).

## 2. The Role of Microenvironment and Immunology

It has become increasingly clear that the reciprocal interaction between tumor cells and the tumor microenvironment plays an essential role in the development of myeloma. In solid tumors, the establishment of tumor-associated stroma facilitates not only tumor growth and progression but also invasive and metastatic growth [3]. Similar to healthy plasma cells, myeloma cells initially depend on signals from the BMME for their survival [3,4,5,6]. The MGUS-to-MM progression requires multiple genomic events and the establishment of a permissive BMME, although it is generally unclear if the various microenvironmental events are causative or consequences of disease progression [7,8,9]. Furthermore, the mainstay of myeloma therapy includes immunomodulatory drugs (IMiDs) such as thalidomide and its analogs lenalidomide, and pomalidomide [10,11], exerting their function, at least in part, by affecting the bone marrow microenvironment.

The BM is a complex organ consisting of numerous highly specialized cell lineages responsible for different tasks such as blood production, immunity, and skeletal integrity. The BMME has been classically divided into several niches such as the vascular niche, the endosteal niche, and the immune microenvironment, which refers more to a functional compartment of differentiated immune cells located in the BM stroma [12,13].

The interplay between myeloma cells and the bone marrow microenvironment is essential for malignant transformation, treatment, and progression (Figure 2). Many cell types are found in the BMME belonging to one of the niches and could be divided into hematopoietic cells (B cells, T cells, natural killer (NK) cells, myeloid-derived suppressor cells (MDSCs), and osteoclasts) and nonhematopoietic cells such as BM stromal cells, osteoblasts, and endothelial cells.

As an immunological organ, the BM includes a wide range of immature and mature innate and adaptive immune cell types and because of the high metabolic turnover the BM is also densely vascularized with different types of arterioles, capillaries, and sinusoids.

## 3. Immunosuppressive BMME and Immune Exhaustion

The progressive transformation of an asymptomatic MGUS into active myeloma is not only accompanied by an increased mutational load but also by significant changes in the cellular composition of the BMME. The subsequent loss of functional immune surveillance leads to BMME-induced immune exhaustion and suppression [14,15]. These dysfunctional cellular compositions involve the recruitment of various immunosuppressive cells, including MDSCs, regulatory T cells (Tregs), regulatory B cells (Bregs), and tumor-associated macrophages (TAMs) in the BMME.

MDSCs are a heterogeneous population of immature myeloid cells, which normally differentiate into macrophages, granulocytes, or dendritic cells and frequently increase during the development of myeloma, peaking in relapsed and/or refractory myeloma patients [16]. It was demonstrated that cancer patients with a high number of MDSCs have a shorter survival compared to patients with a lower MDSC-level. Driven by the activation of the STAT3 pathway that in turn is stimulated through various cytokines such as interleukin 6 (IL-6) or vascular endothelial growth factor (VEGF), MDSCs are able to suppress immune responses by the secretion of nitric oxide (NO), reactive oxygen species (ROS), or prostaglandin E2 and these immune-suppressive cytokines inhibit the proliferation and expansion of Th1 cells, cytotoxic T lymphocytes (CTLs), and NK cells, thereby enabling the differentiation and recruitment of Th17 cells, Tregs, and TAMs to the microenvironment [16,17].

Tregs are one of the most pronounced immunosuppressive cell populations in myeloma. Tregs inhibit Th1, Th17, CTL, macrophage, and dendritic cell (DC) function by direct cellular interactions and via secretion of suppressive cytokines, such as transforming growth factor-beta (TGF-β) and IL-10 [18].

Analogous to MDSCs, the frequency of Tregs gradually increases in the BMME during the progression of MGUS to active myeloma [19], and, conversely, decreases after successful treatment with lenalidomide plus dexamethasone [14,19]. The activation of the STAT3 pathway is crucial for the development, proliferation, and function of Tregs, which is achieved through upregulation of transcriptional expression of forkhead box P3 (*FOXP3*) [20]. Most importantly, a common feature of all immunosuppressive cells, including Bregs and TAMs, is the expression of high levels of CD38, which can be targeted by CD38-directed antibodies such as daratumumab and isatuximab [21,22]. Treatment with daratumumab can rapidly deplete CD38+ Tregs, MDSCs, and Bregs and is associated with clonal expansion of CD4+ and CD8+ T cells in myeloma patients (Figure 3). Hence, CD38-directed antibody therapy—besides targeting CD38-positive myeloma cells—can also restore an immunologically functional BMME exerting appropriate anti-MM T-cell responses [23].

Mesenchymal stromal cells (MSCs) and osteoclasts also contribute to an immunosuppressive environment. Crosstalk between myeloma cells and MSCs mediated by toll-like receptor 4 (TRL4) signaling promote tumor microenvironment transformation and drives MSCs into a phenotype promoting tumor growth and immune escape [14,24]. MSCs in the myeloma BMME suppress T cell activation and proliferation, impair DC maturation, and induce Tregs via the secretion of several cytokines and interleukins such as IL-6, TGF-β, IL-10, and upregulation of surface molecules such as VCAM-1, ICAM-1, or CD40 [14,24,25]. Osteoclasts contribute to an immunosuppressive environment by the production of A proliferation-inducing ligand (APRIL), which is the ligand for B cell maturation antigen (BCMA) that is expressed on virtually all myeloma cells, and for transmembrane activator and calcium modulator and cyclophilin ligand interactor (TACI) that is expressed on myeloma cells and Tregs. APRIL facilitates myeloma cell growth and survival and stimulates the upregulation of TGF-β and IL-10, thereby promoting the survival of Tregs via TACI signaling [26].

As with various solid tumors and hematological cancers, immune-suppressive BMME involves a significant upregulation of immune checkpoint molecules during the transition of MGUS to myeloma. CD8+ T cells from myeloma patients express multiple immune checkpoint receptors, including programmed cell death protein 1 (PD-1), cytotoxic T-lymphocyte-associated protein 4 (CTLA-4), T cell immunoglobulin mucin-3 (TIM-3), lymphocyte activation gene 3 (LAG-3), and, recently, T cell immunoglobulin and ITIM domain (TIGIT) [27,28]. Terminal T cell exhaustion is associated with the loss of cytotoxicity by CD4+ and CD8+ T cells subsets that produce IFN-γ, a critical cytokine for tumor immunity [27]. The expression of immune checkpoint molecule PD-1 on effector T and NK cells, and its ligands PD-L1/2 on myeloma cells is a well-described phenomena induced by an immune-mediated interferon-γ (IFN-γ) response [29]. The PD-L1/2 expression on MM cells can also be enhanced through the stimulation of TLR ligands, interactions with MSCs, or APRIL signaling [14,29]. Recently, the progression of myeloma has been associated with high levels of TIGIT expression on CD8+ T cells, which exhibit impaired proliferative and cytokine responses upon antigen stimulation [30]. Although immune exhaustion seems to play an important role in the pathogenesis of myeloma; monotherapy with antibodies blocking immune checkpoints in relapsed and/or refractory myeloma patients was not effective [31]. Moreover, a randomized phase 3 trial including a checkpoint inhibitor combined with an immunomodulatory substance and dexamethasone for patients with relapsed or refractory multiple myeloma was unsuccessful, raising questions about future targeting of PD-1 and PD-L1 in myeloma [32,33].

## 4. From Bench to Bedside: Therapeutical Targeting of the BMME in MM

During the last years several BBME therapeutical approaches, which modes of action are summarized in Figure 3, have been developed. Furthermore, these approaches are described in the following sections.

### 4.1. Anti-CD38 Therapies

CD38 is a transmembrane glycoprotein expressed on plasma cells and other hematopoietic cells. The introduction of anti-CD38 monoclonal antibodies has changed the treatment landscape of MM in the past years (Figure 3). Daratumumab and isatuximab are approved in different combinations and lines of therapy and are broadly used in myeloma treatment (Table 1) [34,35]. Two trials of combination therapy including the anti-CD38 monoclonal antibody daratumumab, set the new standard of care for non-transplant eligible newly diagnosed multiple myeloma (NDMM).

The phase III ALCYONE trial randomized non-transplant eligible NDMM patients (age >65 years old or exhibiting comorbidities) between standard VMP or daratumumab (Dara)-VMP (bortezomib, melphalan, and prednisone) treatment. A total of 706 patients were enrolled and the rates of overall response (ORR), very good partial response (VGPR), complete response (CR) as well as minimal residual disease (MRD) negativity were significantly higher in the Dara-VMP arm. With more than 3 years of follow-up, Dara-VMP even prolonged overall survival (OS) in patients with newly diagnosed multiple myeloma ineligible for autologous stem cell transplantation (ASCT) and therefore, the combination of daratumumab with VMP was approved by the Food and Drug Administration (FDA) in May 2018 and thereafter, by the European Medicines Agency (EMA) [36,37].

In the randomized phase III M trial, non-transplant eligible NDMM patients received either lenalidomide and dexamethasone (Rd) or Dara-Rd. The rates of ORR, VGPR, CR, and MRD-negativity (evaluated by NGS, threshold of 1 tumor cell per 10^5^ white cells) were significantly higher in the Dara-Rd arm [38]. In a recent update conducted after 48 months of follow-up, the addition of daratumumab to Rd continued to demonstrate a significant PFS benefit, with an estimated 48-month PFS rate of 60% with Dara-Rd vs. 38% with Rd. More patients continued to have deeper and more durable responses with Dara-Rd vs. Rd alone [39].

The CASSIOPEIA study in transplant eligible NDMM patients investigated the efficacy of adding daratumumab to VTD (bortezomib, thalidomide, and dexamethasone) induction and post ASCT consolidation therapies. Overall, 1085 patients were included and received four induction cycles of VTD with or without daratumumab, high dose therapy followed by ASCT, and two additional cycles of consolidation (Dara-VTD or VTD). Patients who responded underwent a second randomization to either daratumumab maintenance or observation. The ORR after consolidation was only slightly higher in the daratumumab arm (92.2% vs. 89.9%). However, rates of MRD negativity (64% vs. 44%; *p* < 0.0001) and stringent CR were significantly better in the D-VTD arm. The addition of daratumumab to VTD improved PFS (HR 0.47; *p* < 0.0001) which was observed across all subgroups of patients [35,40]. CASSIOPEIA was the first study that demonstrated the clinical benefit of adding daratumumab to standard of care in transplant eligible NDMM.

In the relapsed/refractory setting (RRMM), the POLLUX and CASTOR trials compared Dara-Rd and Dara-Vd to Rd and Vd alone, respectively. The POLLUX trial included 569 patients with a median of one previous line of therapy. Rd was administered with or without daratumumab until progression or intolerance. Dara-Rd showed an exceptional ORR of 93% in RRMM, with 55% of patients reaching CR or better. The median PFS was 44.5 months in the Dara-Rd arm and 17.5 months in the Rd arm [41]. Dara-Rd reduced the risk of progression or death by 56%. The benefit was seen across all patient subgroups, including high-risk patients [35]. In the phase 3 CASTOR study, RRMM patients were randomized to Dara-Vd or Vd alone. Vd was discontinued after eight cycles and daratumumab was continued until progression [42]. The ORR, as well as the VGPR, CR, and MRD negativity rates, were significantly better in the Dara-Vd group. The median PFS was 16.7 months in the Dara-Vd arm compared with 7.1 months in the Vd arm (*p* < 0.0001). Daratumumab was also superior in the high-risk cytogenetic group (median PFS 11.2 vs. 7.2 months; HR, 0.45; *p* = 0.0053).

Finally, the combination of daratumumab and carfilzomib was studied in the phase 3 CANDOR trial evaluating carfilzomib and dexamethasone (Kd) with or without daratumumab (Dara-Kd). Four hundred and sixty-six patients were included. The ORR was 84.3% in the Dara-Kd arm vs. 74.7% in the Kd arm, and, particularly, the CR (28.5% vs. 10.4%) and MRD negativity (17.6% vs. 3.9%; threshold 10^−5^) rates were significantly better in the Dara-Kd arm. The median PFS was 28.6 months in the Dara-Kd group vs. 15.2 months in the KD group (HR 0.59) [35,43].

Isatuximab is another anti-CD38 IgG kappa monoclonal antibody with a similar mechanism of action as daratumumab and is characterized by a strong anti-myeloma activity via direct tumor cell killing and a unique direct proapoptotic effect independent of Fc crosslinking [44].

In the IKEMA trial, RRMM patients were randomly assigned to isatuximab plus carfilzomib–dexamethasone (Isa-Kd) or carfilzomib–dexamethasone (Kd, control group). Treatment was continued until progression or unacceptable toxicity. The study included 302 RRMM patients with one to three prior treatment lines. Median PFS was not reached in the Isa-Kd group vs. 19.2 months in the Kd group (HR 0.531; *p* = 0.0007), representing a 47% reduction in the risk of disease progression or death. The benefit was seen across all subgroups. ORR was 86.6% in the Isa-Kd arm vs. 82.9% in the Kd arm. Additionally, CR (39.7% vs. 27.6%) and MRD negativity (threshold 10^−5^; 29.6% vs. 13.0%) rates were better in the Isa-Kd arm [35,45].

The phase III ICARIA trial compared the triplet Isa-Pd (isatuximab, pomalidomide, and dexamethasone) to Pd in 307 patients with RRMM who had received at least two previous lines of therapy (with a median of three lines). After a median follow-up of 11.6 months, a benefit in terms of ORR (60% vs. 35%; *p* < 0.001) and PFS (median PFS 11.5 vs. 6.5 months; HR 0.59; *p* = 0.001) for the triplet arm was observed. The PFS advantage was not statistically different in high-risk patients (median PFS 7.5 vs. 3.7 months; HR 0.66, 95% CI 0.30–1.28) [44,46].

### 4.2. Immunomodulatory Substances (IMiDs)

The outcomes of patients with MM have improved substantially. One reason is the development of the immunomodulatory drugs (IMiDs), which include thalidomide, lenalidomide, and pomalidomide (Figure 3). Although thalidomide is now less commonly prescribed because of well-known side effects such as polyneuropathy, it is still used within the CASSIOPEIA schedule in transplant-eligible NDMM patients, and has been designated as an IA indication in recent ESMO guidelines [47]. Lenalidomide is more widely used than thalidomide and is approved for treating transplant-eligible and transplant ineligible NDMM patients. Lenalidomide is also included in many combination triplet schemes in the RRMM setting and is also the standard of care in post-transplant maintenance. Pomalidomide, on the other hand, is currently only approved in the RRMM setting. IMiDs have been described to exert a multitude of functions including anti-angiogenic, cytotoxic, and immunomodulatory ones. On a molecular basis, recent publications report that their mechanism of action is based on binding cereblon and thus regulating the ubiquitination of key transcription factors including Ikaros and Aiolos [48].

In the first-line setting, lenalidomide combined with bortezomib and dexamethasone (VRd) is still one of the most commonly used triplet regimens in transplant eligible (IIB) and transplant non-eligible (IA) NDMM patients and is recommended for MM front-line therapy in the recent ESMO guidelines [47] based on the SWOG S0777 trial comparing VRd vs. Rd in patients with previously untreated MM without an intent for immediate ASCT [49]. This updated analysis included 460 patients. After a median follow-up of 84 months, the median PFS was 41 months for VRd and 29 months for Rd. The addition of bortezomib to lenalidomide and dexamethasone for induction therapy resulted in a statistically significant and clinically meaningful improvement in PFS and a better OS.

In transplant-eligible NDMM, maintenance with lenalidomide is still considered the standard of care for all myeloma patients post-ASCT [47]. Lenalidomide maintenance after ASCT offers PFS and OS benefits over placebo, as reported in two large randomized trials [50,51] and a meta-analysis including more than 1200 patients, with a median follow-up of 79.5 months [52]. Compared to placebo, lenalidomide maintenance offered more than 2 years of PFS (52.8 vs. 23.5 months) and 2.5 years of OS benefit [52].

In RRMM, lenalidomide-sensitive patients in whom a salvage ASCT is not considered, based on both HR and absolute values of median PFS, Dara-Rd (POLLUX regimen) provides the longest PFS for patients with RRMM with one to three prior lines of therapy and standard-risk cytogenetic profile. However, another potent triplet is the combination of the most efficient proteinase inhibitor carfilzomib with Rd (KRd). KRd has shown a significant OS benefit over Rd: median OS 48.3 months vs. 40.4 months for KRd vs. Rd (HR 0.79; *p* = 0.0045), respectively [53].

However, since most patients progressing after first-line therapy are considered lenalidomide-refractory these days, they should not receive lenalidomide in any further treatment line. Instead, such lenalidomide-refractory patients could receive either Dara-Vd (CASTOR), or combinations of daratumumab or isatuximab with carfilzomib and dexamethasone (Dara-Kd or Isa-Kd) as new standard options for this setting. A further option is combining Pomalidomide with Vd (PVd) (OPTIMISSM study). This triplet was compared with Vd in RRMM patients who had received one to three prior lines of therapy that included lenalidomide. More than 70% of the patients were refractory to lenalidomide. After a median follow-up of 16 months PVd significantly improved PFS compared with Vd (median 11.2 months vs. 7.1 months; HR 0·61; *p* < 0·0001) [54] and supports the use of PVd in lenalidomide-refractory RRMM patients.

In conclusion, IMiDs exhibit pleiotropic effects directly targeting myeloma cells and enhancing the immune response in the immuno-suppressive BMME. Further, their synergistic potential with monoclonal antibodies, e.g., daratumumab or isatuximab, represents a novel treatment standard and has significantly improved the long-term survival of myeloma patients.

### 4.3. CAR T Cells

Chimeric antigen receptor (CAR) T cells are genetically modified to target antigen-expressing tumor cells. CAR T cells express chimeric proteins that consist of extracellular tumor antigen recognition domains along with intracellular T cell receptor effector domains and co-stimulatory molecules. This results in tumor cell recognition and subsequent activation of CAR T cells. The main advantages of CAR T cells for cancer treatment are the following: (1) CAR T cells recognize their native targets without the need for prior antigen processing and human leukocyte antigen (HLA)-dependent antigen presentation. Importantly, this allows them to effectively act even under immunosuppressing conditions. (2) They can be used as a polyclonal T-cell population against a selected antigen represented on the surface of tumor cells, consequently augmenting tumor-targeted T-cell numbers [55,56,57]. To date, CAR T cells have been successfully tested in a broad range of tumor settings. CAR T cell therapy [58,59] is currently approved by the FDA and EMA to treat aggressive B-cell lymphomas and achieved impressive response and remission rates even in refractory stage [60,61,62]. However, CAR T cells possess, besides their great anti-tumoral capacity, severe side effects reported especially for B-cell malignancies. Among them are B-cell aplasia, cytokine release syndrome (CRS), macrophage activation syndrome, and neurotoxicity [63,64,65].

The first CAR T cells targeting BCMA, also known as TNFRSF17 or CD296, were pre-clinically tested in 2013 on myeloma cell lines and showed a marked activity against them (Figure 3) [66]. Since then, several new anti-BCMA CAR T-cell constructs have been developed and are explored mainly in RRMM patients, as summarized in Table 2. The expression of BCMA was reported to be restricted to the B-cell lineage, especially on the surface of plasmablasts, differentiated plasma cells, and malignant myeloma cells, but not on hematopoietic stem cells, naïve B, and memory B cells. Two distinct ligands, B-cell activating factor (BAFF) and a proliferation-inducing ligand (APRIL), bind to the BCMA and are responsible to support the long-term survival of B cells during differentiation [67,68,69]. However, it was recently reported that BCMA is also involved in the neural development; thus, it is speculated that neurotoxicity with BCMA-directed CAR-T therapies, might be a direct on-target neurotoxicity [70]. To date, idecabtagene vicleucel (ide-cel, bb2121) and ciltacabtagene autoleucel (cilta-cel) are the only approved CAR T-cell products to treat RRMM. In addition to ongoing clinical trials with anti-BCMA CAR T cells, CAR T cells targeting CD38, CD138, and SLAMF7 are under investigation.

Another interesting approach of CAR T-cell therapy in MM is the use of allogenic T cells in the UNIVERSAL study, in which only a small number of patients were treated. In this study, the allogenic T cells were genetically modified to express the CAR construct on the one hand and to disrupt the TCR constant region preventing the graft-versus-host-disease (GVHD) on the other hand. The allogenic CAR T cells were well tolerated at all tested doses. Remarkably, no GVHD and neurotoxicity were described although high response rates were detected. The great advantage of this technique is the rapid availability of the CAR T-cell product [71,72,73].

### 4.4. Bispecific Antibodies

A promising novel alternative possibility to redirect T cells to tumor cells is the strategy to use bispecific antibodies (BsAbs) because these antibodies are available off-shelf and are relatively easy to administer (Figure 3). The BsAbs activate T cells by binding CD3ε of the T-cell receptor complex leading to T-cell activation, which is independent of the major histocompatibility complex (MHC) restriction [86,87]. Furthermore, BsAbs possess the ability of T-cell activation in the absence of co-stimulatory signals, finally resulting in T-cell-mediated tumor cell lysis [88,89].

Based on the presence or absence of an Fc domain, two distinct groups of BsAbs can be distinguished. Fc domain-containing BsAbs induce additional immune responses mediated by innate immune cells and/or the complement system. Antibodies without an Fc region consist of two different single-chain variable regions and are known as bispecific T-cell engagers (BiTEs) (Figure 3) [90].

The main toxicities of BsAbs are CRS and neurotoxicity—although occurring to a lesser extent and severity than CART cell therapy—mainly during the step-up dosing or during the first administrations. Further side effects include cytopenia and infections [91]. Another challenging fact is T-cell exhaustion, antigen escape, or an immunosuppressive microenvironment resulting ultimately in tumor cell resistance. However, different strategies to overcome these problems have been developed, i.e., a combination of BsABs with immunomodulatory drugs [92,93].

To date, more than 10 different BsAbs, mostly targeting CD3 and BCMA, are under clinical investigation with promising results (Table 3).

### 4.5. Antibody-Drug Conjugates Targeting BCMA

One of the approaches targeting BCMA involves the use of an antibody-drug conjugate (ADC), which is a combination of (i) a monoclonal antibody (mAb), (ii) a cytotoxic agent, and (iii) a linker joining these two elements. The mAb provides selective delivery of ADCs to target cells by binding to specific cell surface molecules and inducing cell death after internalization of the ADC (Figure 3) [69,108].

SG1-vcMMAF8 containing the drug monomethyl auristatin F (MMAF) was the first potent ADC targeting BCMA and demonstrated cytotoxic activity against BCMA-positive myeloma cell lines [109].

Belantamab mafodotin (GSK2857916; belamaf), an afucosylated humanized mAb conjugated to MMAF that inhibits tubulin polymerase, was the first anti-BCMA ADC to be tested in clinical trials [110]. The therapeutic effect of this ADC is achieved through three distinct mechanisms of action: (i) antibody-dependent cytotoxicity (ADCC), which is enhanced by increased binding of the afucosylated domain of Fc to FcgR (FcgRIIIa) expressed on effector cells, (ii) induction of apoptosis by disruption of microtubules by MAAF, and (iii) prolonged stability of ADC in the blood and reduced killing of non-target cells, provided by the uncleavable maleimidocaproyl linker [110]. Based on the results from DREAMM-2 (NCT 03525678) multicenter trial, which showed an overall response rate of 31% [111], the U.S. Food and Drug Administration (FDA) approved belantamab mafodotin in August 2020 as monotherapy for adult patients with RRMM who have been treated with at least four prior therapies, including an anti-CD38 mAb, a proteasome inhibitor (PI) and an immunomodulatory drug.

MEDI2228 is another ADC targeting BCMA, consisting of a fully human anti-BCMA antibody conjugated by a protease removable linker to a DNA cross-linking pyrrolobenzodiazepine dimer (PBD). Results from a phase 1, first-in-human, open-label, dose-escalation trial (NCT03489525) of MEDI2228 in patients with RRMM demonstrated clinical efficacy at all dose levels tested. MEDI2228 at 0.14 mg/kg administered intravenously every three weeks had a safe profile and an objective response rate of 61% in the RRMM patient population heavily pretreated with PIs, IMiDs, and mAbs [112].

HDP-101 is an anti-BCMA ADC that has been tested in preclinical MM models. It exhibits cytotoxicity regardless of the proliferation state of tumor cells through the use of alpha-amanitin, an inhibitor of eukaryotic RNA polymerase II. HDP-101 showed high efficacy against both primary MM cells from patients with newly diagnosed and RRMM in vitro, induced tumor regression in mouse xenograft models, and was well tolerated with a promising therapeutic index in cynomolgus monkeys [113].

### 4.6. BMME Mediated Therapy Resistance

Since most of the BMME targeting therapies, described in Section 4.1, Section 4.2, Section 4.3, Section 4.4 and Section 4.5, restore/activate an anti-MM immune response, it is observed that the BMME plays a key role in therapy resistance for these drugs.

One possibility could be the recruitment of immunosuppressive cells, including MDSCs, Tregs, Bregs, and TAM into the BMME of MM. These immunosuppressive cells might secrete NO, arginase, ROS, prostaglandin E2 (PGE2), or indoleamine 2,3-dioxygenase (IDO) [14,24,25] as well as of immunosuppressive cytokine (e.g., IL-10 and TGF-β) inhibiting the proliferation and expansion of Th1 cells, CTLs, and NK cells [15,16]. Additionally, it has also been demonstrated that MSCs exert immunomodulatory properties, which can be mediated by secretion of several factors, including IL-6, TGF-β, IL-10, PGE2, and upregulated expression of several surface molecules such as VCAM-1, ICAM-1, and CD40 also resulting in immunosuppressive conditions [24,25,114,115,116,117]. Another possibility for BMME targeting therapy resistance might be caused by the cross-talk of BMME and MM cells via soluble factors such as IL-6, APRIL, and growth factors, but most importantly via the integrin-mediated cell adhesion and Notch signaling resulting in inhibition of apoptosis [118,119,120,121], on which the major cytotoxic machinery of the immune cells significantly depends [122,123,124,125].

For therapeutic approaches employing CAR T cells as well as bispecific antibodies, it was observed that MSCs could protect MM cells from highly lytic BCMA-CAR T-cells [126], whereas the lytic function of BCMA/CD3 bispecific antibodies was not that much influenced by MSCs [127].

## 5. Conclusions

The interaction between tumor cells and the tumor microenvironment plays a crucial role in the stepwise development of myeloma from MGUS to active MM. Furthermore, the establishment of a tumor-promoting stroma facilitates tumor growth and myeloma progression. Since myeloma cells depend on signals from the microenvironment for their survival and growth, therapeutic strategies targeting the microenvironment represent novel therapeutic approaches to treat MM patients. At present, these novel treatment options are often allocated in the later course of disease when most of the patients are already heavily pretreated and display a dysfunctional/compromised immune system. Treatment efficacy may be increased by bringing these new agents into the earlier lines of therapy.

Over the past few years, a number of novel treatment strategies have been developed, such as bispecific antibodies that recognize both BCMA and CD3ε (allogeneic), CAR T cells, ADCs, and anti-CD38 antibodies or second-generation IMiDs, and proteasome inhibitors which have significantly improved patient outcomes. However, there is a lack of clear consensus on the use, combination, and sequence of these new therapies and finally, most myeloma patients will experience relapse due to the development of resistance mechanisms because of continuous treatment with (novel) agents. Understanding the pathophysiology of resistance to commonly used anti-myeloma agents requires further research. Furthermore, new targets in the myeloma cells or in the BMME have to be identified to develop therapeutical approaches with a novel mode of action, which might help to overcome this drug resistance.

## Figures and Tables

**Figure 1 ijms-23-07627-f001:**
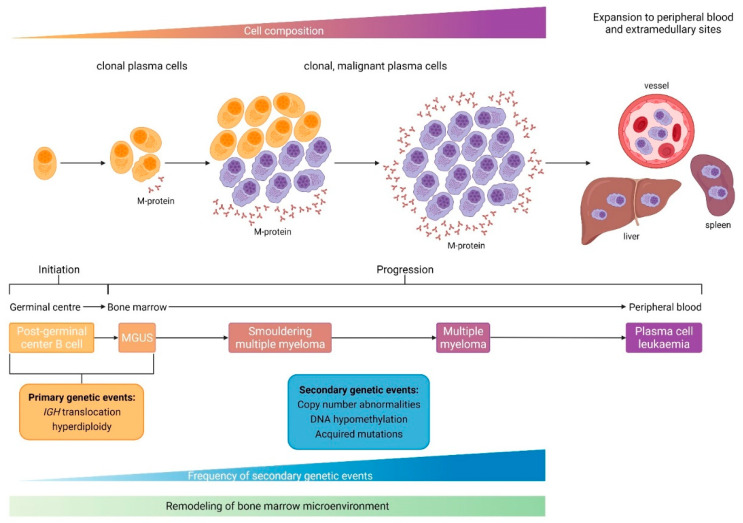
The development process of monoclonal gammopathies. Multiple myeloma arises as a result of complex and multistep changes in the bone marrow and is preceded by precursor states such as monoclonal gammopathy of undetermined significance (MGUS) and smoldering multiple myeloma (SMM). Adapted from Kumar et al. [1]. Created with BioRender.com (accessed on 30 June 2022).

**Figure 2 ijms-23-07627-f002:**
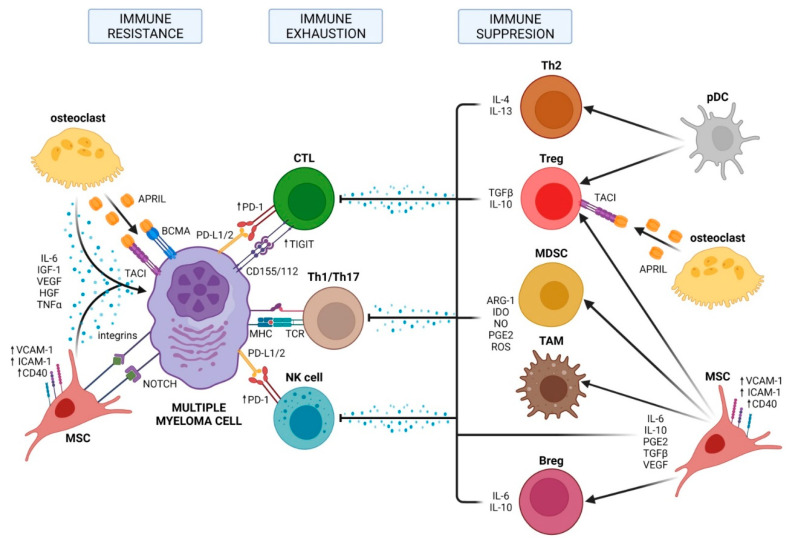
Three major mechanisms of immune evasion mediated by bone marrow microenvironment in MM: immune resistance, immune exhaustion, and immune suppression. Adapted from Holthof et al. [14]. For explanation see text. Created with BioRender.com (accessed on 30 June 2022).

**Figure 3 ijms-23-07627-f003:**
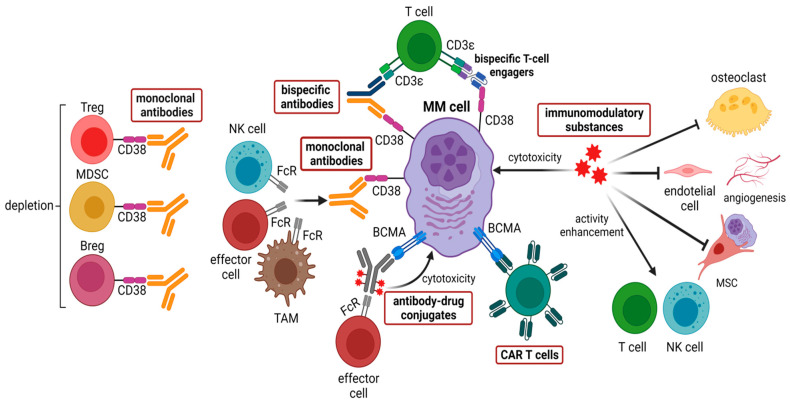
Therapeutic approaches in multiple myeloma targeting the bone marrow microenvironment. Among these, five main types can be distinguished: (i) monoclonal antibodies targeting CD38 such as daratumumab and isatuximab, (ii) immunomodulatory substances such as thalidomide, lenalidomide, and pomalidomide, (iii) CAR T cells target mainly BCMA, other targets such as CD38, CD138, and SLAMF7 are being developed, (iv) bispecific antibodies bind mainly CD3 and BCMA, which are currently under clinical investigation, (v) antibody-drug conjugates, including the FDA-approved belantamab mafodotin along with MEDI2228 and HDP-101, which are still being tested. Created with BioRender.com (accessed on 30 June 2022).

**Table 1 ijms-23-07627-t001:** Selected trials of daratumumab and isatuximab in RRMM.

Study Name	Phase	Setting	Treatment	Number of Patients	ORR (%)	CR (%)	MRD Neg (%)	NCT Number	References
ALCYONE (MMY3007)	3	NTE NDMM	Dara-VMP vs. VMP	706	90.9 vs. 73.9	42.6 vs. 24.4	22.3 vs. 6.2	NCT03158688	[36,37]
MAIA (MMY3008)	3	NTE NDMM	Dara-Rd vs. Rd	737	92.9 vs. 81.3	47.6 vs. 24.9	24.2 vs. 7.3	NTC02252172	[38,39]
CASSIOPEIA (MMY3006)	3	TE NDMM	Dara-VTD vs. VTD	1085	92.6 vs. 89.9	39 vs. 26	64 vs. 44	NTC02541383	[35,40]
POLLUX (MMY3003)	3	RRMM	Dara-Rd vs. Rd	569	92.9 vs. 76.4	43.1 vs. 19.2	22.4 vs. 4.6	NCT02076009	[41]
CASTOR (MMY3004)	3	RRMM	Dara-Vd vs. Vd	500	85 vs. 63 92 vs. 74 (1 prior line treatment)	30 vs. 10 43 vs. 15	14 vs. 2 20 vs. 3	NCT02136134	[42]
CANDOR	3	RRMM	Dara-Kd vs. Kd	466	84.3 vs. 74.7	29 vs. 10	14 vs. 3	NCT03158688	[35,43]
IKEMA	3	RRMM	Isa-Kd vs. Kd	302	87 vs. 83	40 vs. 28	30 vs. 13	NCT03275285	[35,45]
ICARIA-MM	3	RRMM	Isa-Pd vs. Pd	307	60 vs. 35	5 vs. 2	5 vs. 0	NCT02990338	[44,46]

NTE = non transplant eligible, TE = transplant eligible, RRMM = relapsed/refractory multiple myeloma, ORR = overall response rate, CR = complete response, and MRD = minimal residual disease.

**Table 2 ijms-23-07627-t002:** Characteristics, efficacy, and safety data from selected clinical trials of anti-BCMA CAR T cell constructs in RRMM.

CAR Construct	Study Name	Antigen	Number of Patients	High Risk/EMD (%)	CR (%)	CRS (%)	Neurotoxicity (%)	NCT Number	References
Ide-cel (bb2121)	CRB-401, Phase 1	BCMA	62	27/37	39	76	36	NTC02658929	[71]
Ide-cel (bb2121)	KarMMa, Phase 2	BCMA	128	35/39	33	84	18	NCT03361748	[74,75]
Cilta-cel (LCAR-B38M)	LEGEND-2, Phase 1/2	Biepitope to BCMA (VHH1 and VHH2)	57	NR	74	90	1	NCT03090659	[76]
Cilta-cel (JNJ-4528)	CARTITUDE-1, Phase 1b/2	Biepitope to BCMA (VHH1 and VHH2)	97	24/13	80	95	21	NCT03548207	[77,78]
Orva-cel (JCARH125)	EVOLVE, Phase 1/2	BCMA	62	41/23	36	89	13	NCT03430011	[79]
bb21217	CRB-402, Phase 1	BCMA	69	33/NR	29	70	16	NCT03274219	[80]
CAR-BCMA	Phase 1	BCMA	24	46/NR	8	71	NR	NCT02215967	[81,82]
UPenn-CART-BCMA	Phase 1	BCMA	25	96/28	8	88	32	NCT02546167	[83]
CT053	LUMMICAR STUDY 2, Phase 1b/2	BCMA	20	55/25	25	79	16	NCT03915184	[84]
ALLO-715	UNIVERSAL, Phase 1	BCMA	31	48/23	VGPR: 40	45	0	NCT04093596	[73]
C-CAR088	Phase 1	BCMA	23	81/NR	44	91	4	NCT03751293 NCT03815383 NCT04322292 NCT048295018	[85]

CAR = chimeric antigen receptor, EMD = extra-medullary disease, CR = complete remission, CRS = cytokine release syndrome, NR = not reported, VGPR = very good partial remission, and BCMA = B cell maturation antigen.

**Table 3 ijms-23-07627-t003:** Characteristics, efficacy, and safety data of bispecific antibodies in RRMM.

Agents	Type	Phase	Target	Number of Patients	ORR (%)	CRS (%)	NCT Number	References
AMG420	BiTE	1	BCMAxCD3	42	70	38	NCT03836053	[94]
AMG701	BiTE	1/2	BCMAxCD3	75	83	61	NCT03287908	[95]
Teclistamab (JNJ-64007957)	BsAb	1/2	BCMAxCD3	149	69	55	NCT04557098 NCT03145181	[96,97,98]
REGN5458	BsAb	1/2	BCMAxCD3	49	62.5	39	NCT03761108	[99]
TNB-383B	BsAb	1	BCMAxCD3	58	80	45	NCT03933735	[100]
Elranatamab (PF-06863135)	BsAb	2	BCMAxCD3	30	83.3	73	NCT04649359	[101]
CC-93269	BsAb	1	BCMAxCD3	30	89	77	NCT03486067	[102]
GBR1342	BiTE	1	CD38xCD3	19	NR	NR	NCT03309111	[103]
AMG424	BsAb	1	CD38xCD3	NR	NR	NR	NCT03445663	[104]
Talquetamab (JNJ-64407564)	BsAbs	1	GPRC5dxCD3	NR	NR	NR	NCT04108195 NCT03399799 NCT04773522	[105,106]
Cevostamab (BFCR4350A)	BiTE	1	FCRH5xCD3	160	54.8%	80.7%	NCT03275103	[107]

ORR = overall response rate and CRS = cytokine release syndrome.

## Data Availability

Not applicable.

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
