# Peer review of "Targeting the Microenvironment for Treating Multiple Myeloma"

_ijms, 2022, doi:10.3390/ijms23147627_

Round 1
Reviewer 1 Report
The review perfectly summarizes the current state of knowledge in the field of multiple myeloma microenvironment. Reported clinical trials are well summarized with key informations and results being highlighted.
The document is well structured, clear, easy to follow and the figures are very well illustrated.
The legends are complete. However the text in Figure 1 is difficult to read. Increasing the font size by at least one point would make it easier to read.
In section 4.1 Anti-CD38 therapies, a table summarizing the different clinical trials described in this section would be very helpful, as the authors have done for section 4.3 CAR T cells and 4.4 Bispecific antibodies.
Author Response
Thank you for the detailed examination of our work, the positive remarks, and the suggestions for improvement.
Based on the addressed points, we now increased the size of figure 1 and added a table summarizing the different clinical trials in section 4.1.
Reviewer 2 Report
The present review "Targeting the microenvironment for treating multiple myeloma" by Neumeister et al. reports a comprehensive and exaustive state of the art on the several possible therapeutic strategies, available to date, to treat mulptiple myeloma. Those reported strategies are based on the purpose to target the tumor microenvironment since a complex interplay between myeloma cells and the tumor microenvironment occurs and this link is well described in the present report.
Author Response
We thank the reviewer for the positive evaluation of your submitted manuscript.
Reviewer 3 Report
The presented review summarizes a highly relevant topic with a lot of provided information. In my opinion, it is a nice, comprehensive, and up to date review of relevant literature. The authors globally summarized the composition of tumor microenvironment in multiple myeloma and the corresponding immunotherapies. Up-to-date clinical trials have been summarized in this manuscript. This article is well-organized and below listed my comments:
1. This is an interesting manuscript, on some points a bit too general, owed to the huge field of immunotherapies. At some points the review could be more relevant by including detailed examples of how TME was affected in discussed immunotherapies. As the previous section introduced the composition of TME. It would be more comprehensive to involve the influence on those immune cells such as cytokine secretion to modulate TME.
2. To help readers easy to follow and keep focus, I strongly suggest the authors to include a few more figures to illustrate mechanism-of-action of corresponding immunotherapies.
3. As there are limitations in current developing treatments, I’d like to see more discussions in future directions to overcome problems in current strategies.
Author Response
We thank the editor for the detailed examination of our submitted paper and for the comments.
Reviewer 4 Report
Comprehensive and well presented overview aimed at clinicians. Those with an interest in molecular mechanisms will seek relevant informations in recent reviews on these subjects.
Note: Authors in reference #47 needs editing
Author Response
We thank the reviewer for the detailed examination of our submitted article and for pointing out the need for changes of the reference list, which has been done.